# Recent Progress on Tick-Borne Animal Diseases of Veterinary and Public Health Significance in China

**DOI:** 10.3390/v14020355

**Published:** 2022-02-09

**Authors:** Weijuan Jia, Si Chen, Shanshan Chi, Yunjiang He, Linzhu Ren, Xueli Wang

**Affiliations:** 1College of Animal Science and Technology, Inner Mongolia Minzu University, Tongliao 028000, China; jiawj199696@163.com (W.J.); chiss1997@163.com (S.C.); 13296913393@163.com (Y.H.); 2College of Animal Sciences, Key Lab for Zoonoses Research, Ministry of Education, Jilin University, Changchun 130062, China; chensi_1024@163.com

**Keywords:** tick, bacteria, viruses, protozoa, tick-borne diseases

## Abstract

Ticks and tick-borne diseases pose a growing threat to human and animal health, which has brought great losses to livestock production. With the continuous expansion of human activities and the development of natural resources, there are more and more opportunities for humans to contract ticks and tick-borne pathogens. Therefore, research on ticks and tick-borne diseases is of great significance. This paper reviews recent progress on tick-borne bacterial diseases, viral diseases, and parasitic diseases in China, which provides a theoretical foundation for the research of tick-borne diseases.

## 1. Introduction

Ticks are a type of arthropods that are obligately blood-sucking all over the world [1,2]. They often parasitize on the surface of humans and animals, playing important roles as vectors or intermediates for pathogens including bacteria, viruses, and protozoa [3,4,5].

Ticks belong to the class *Acariidae*, order *Ixodidae*, which consists of the families *Ixodidae*, *Argasidae*, and *Nuttalliellidae*. Up to now, more than 800 species of 18 genera have been identified worldwide, while in China, 119 species of 10 genera have been identified, including 100 species of hard ticks and more than 10 species of soft ticks [6,7]. *Ixodesidae* have the most species and are the most harmful, followed by *Argasidae* and only one species of *Nuttalliellidae*. Its body is brown, oblate, like rice grains, and its volume increases like red beans after sucking blood [7]. When not sucking blood, the abdomen is flat and the back is slightly protruding [7]. The development process is metamorphic development, which is divided into four stages: egg, larva, nymph, and adult. All stages except the egg stage need to suck blood, and need to replace 2~3 hosts, and leave after each bloodsucking [8]. 

The distribution of ticks is closely related to the natural environment and has significant regional and seasonal characteristics [9,10,11]. Ticks can be seen throughout the year in tropical regions, while the distribution of ticks in temperate regions is related to the seasons. Furthermore, climatic factors, such as temperature, humidity, and precipitation, all affect the growth, development, and survival rate of ticks [9]. In addition, the increase and decrease in vegetation caused by climate change, as well as the change of type and range of human activities, will also have a certain impact on the distribution of ticks [10]. 

The main harm of ticks is to bite and suck the host’s blood, causing direct damage to them, such as skin allergic reactions, hindering the development of young animals, and reducing the milk production of dairy cows [12]. At the same time, a variety of important pathogens such as bacteria, rickettsia, viruses, and protozoa are indirectly transmitted through saliva, basal ganglia fluid, midgut reflux, and excreta, leading to tick-borne diseases [13,14,15,16,17]. In this review, we discuss recent progress on tick-borne animal diseases in China to provide a theoretical basis for the prevention and control of tick-borne diseases.

## 2. Bacterial Disease

### 2.1. Brucella

Brucellosis is a zoonotic bacterial disease caused by *Brucella* spp., the main symptoms are abortion, sterility, stillbirth, meningitis, subcutaneous abscess [18]. The bacteria have a strong resistance under natural conditions, which can survive in water and soil for a long time and can spread all year round. As a regional disease, the disease is most likely to occur in high temperature and high humidity seasons mainly concentrated in Inner Mongolia and Xinjiang, and other places [19]. Tick obtains *Brucella* by sucking the blood of animals infected by bacteria. The bacteria then colonize in the intestinal tract of the ticks and prevail in the whole development cycle of ticks, including eggs, larvae, nymphs, and adults. Thereafter, the bacterium was transmitted to healthy animals through direct bites of the ticks [20,21].

To cultivate *Brucella* wild-type strains from eggs of *Dermacentor marginatus* (*D. marginatus*) and evaluate the presence of *Brucella* DNA at different developmental stages of *D. marginatu*, Wang Q et al. collected 350 adult female ticks filled with blood from sheep and cattle [22], followed by putting it into clean suitable spawning ventilation tubes to lay eggs and larvae. Then, levels of *Brucella* DNA in female ticks and their progeny were detected based on the *Brucella* outer membrane protein gene 22 (omp22) and IS711 genes. The detection rate of the *Brucella* omp22 gene was 4.6% (16/350) in female ticks and 40.9% (90/220) in larvae, which developed from brucella-positive eggs. Li Y et al. collected 750 blood-sucking female ticks in Xinjiang and found that the detection rate of *Brucella* in the collected ticks was 26.2%, with 96–100% nucleic acid homology to that of the sequences deposited in GenBank [23]. Huang T collected 2256 ticks from 23 pastures in Hulunbuir, Inner Mongolia, and extracted DNA from different developmental stages of ticks to detect *Brucella* [24]. The brucellosis-specific gene Bscp31 was amplified using salivary gland and midgut tissue as templates. The results showed that the predominant tick species was *Dermacentor nuttalliolener* (*D. nuttalliolener*). The positive rate of the specific gene Bscp31 of *Brucella* spp. ranged from 0.00~87.80%, and the highest was 89%. The specific genes of *Brucella* spp. were detected at all stages of ticks.

Jiang et al. collected a total of 747 parasitic ticks and 337 free ticks on the surface of animals in 11 counties/cities along the Xinjiang border [25]. The results showed that the positive rate of *Brucella* was 19.74% (214/1084). The carrying rates of *Brucella* in parasitic ticks and free ticks were 25.30% and 7.42%, respectively. The carrying rate of *Brucella* in parasitic ticks was significantly higher than that of the free ticks (χ^2^ = 46.873, *p* < 0.05).

The above studies show that *Brucella* is widely distributed in parasitic ticks and free ticks, and both *Dermacentor nuttalli* (*D*. *nuttalli*) and *D*. *nuttalliolener* can carry the bacteria. At present, brucellosis is classified as a second-class infectious disease in China, and ticks, as the storage host of *Brucella*, promote the spread of the disease. Therefore, strengthening the tick eradication work in breeding sites can reduce the risk of transmission of the disease, thereby reducing the incidence of the disease.

### 2.2. Tulabacterium

*Tulacobacteria*, also known as *Francisella tulacobacteria*, is a Gram-negative *coccyanobacterium* with a size of about 0.3~0.7 × 0.2 μm, which causes rabbit fever, and is an acute natural epidemic disease [26]. This bacterium can live in low temperatures and water for a long time, and the incubation period is generally 3~5 d. The bacterium is often detected in rabbits and other rodents, especially voles, squirrels, and other small rodents [27]. The disease can be prevalent throughout the year, usually in late spring and early summer, and the epidemic areas that have been found in China are mainly concentrated in Inner Mongolia, Xinjiang, Tibet, and other border areas [28].

Li et al. carried out the detection of *Tularemia* in ticks carried by cattle and sheep in some provinces in China and sequenced the subspecies and genes of *Tularemia* in the positive samples [29]. The results showed that the positive rate of *Tulabacterium* in ticks carried by cattle and sheep was 3.1% (15/490). PCR results show that its subspecies is *Francisella tularensis* subsp. *holarctica*, which belongs to the most virulent subspecies. Wei and colleagues tested 4797 ticks caught in the three northeastern provinces and Inner Mongolia for *Tularemia* and found that the average detection rate of the bacteria was 1.45%, and the total detection rates in Jilin Province and Inner Mongolia were 2.37% and 3.37%, respectively, while both Liaoning Province and Heilongjiang Province were negative [30]. Zhang et al. collected 1670 ticks in two Tularia endemic areas (Inner Mongolia, Heilongjiang) and two non-endemic areas (Jilin, Fujian) and detect the bacteria via nested PCR [31]. The average detection rate of ticks was 1.98%, and the positive rates of different types of ticks were significantly different (*p* < 0.05), mainly *Dermacentor silvarum* (*D. silvarum*) and *Ixodes persulcatus* (*I. persulcatus*). These results indicate that ticks are important intermediates of *Tularemia*. Although the genotype of the bacteria can be identified by PCR, the correlation between ticks and infections of the bacteria in cattle and sheep needs further study.

*Tulabacillus* is the causative agent of Tulabacillosis, which can infect humans and animals by tick-biting, directly contacting infected animals, ingesting contaminated water or food, and inhaling infectious aerosols [32,33]. The main symptoms of the disease are high fever, swollen lymph nodes, fatigue, shortness of breath, etc. In severe cases, death is due to respiratory failure [34]. *Tulabacterium* is highly contagious because it easily forms aerosols and spreads in the air. It is recognized as one of the dangerous biological weapons [35,36]. At present, there is no licensed vaccine against *Tulariaceae*, and the pathogenic mechanism of the bacteria is still not fully known [37]. Therefore, it is very important to prevent the infection and spread of bacteria. In the process of livestock breeding, measures such as killing ticks on the surfaces of livestock houses and animals should be taken to cut off the transmission route of germs and reduce the occurrence of diseases.

### 2.3. Rickettsia

*Rickettsia* is an obligate intracellular parasitic Gram-negative bacterium, which mainly parasitizes on arthropods and causes a variety of zoonotic diseases through tick-borne transmission [38,39,40]. The genus *Rickettsia* is classified into four groups: the *spotted fever group rickettsia* (*SFGR*), the *typhus group rickettsia* (*TGR*), the *ancestral group rickettsia* (*AGR*), and *transitional group rickettsia* (*TRGR*) [41].

*TGR* includes *R. mooseri* and *R. prowazekii*, while *SFGR* is a complex ecological group with many types, including *R. conorii*, *R. martensii*, *R. sibirica*, and so on [42]. *AGR* includes *Rickettsia bellii* (*R. bellii*) and *Rickettsia canadensis* (*R. canadensis*), and *TRGR* contains *Rickettsia felis* (*R. felis*) and *Rickettsia akari* (*R. akari*) [43]. The occurrence of the disease shows obvious seasonality, mostly concentrated in April to September and often occurs in Xinjiang and other places. It is a type of zoonotic disease that seriously threatens human health [44].

Zhou and colleagues collected 320 free ticks in the Qiqian area of Inner Mongolia to detect and genotype the SFGR in the ticks [45]. The results showed that the positive rate of *SFGR* was 47.50% (152/320), and there were three genotypes, namely *Candidatus Rickettsians Tarasevich* (*CRT*), *R. lauticus*, and *R. heilongjiangensis*. Among them, *I. persulcatus* carries 3 species of *rickettsia* with a positive rate of 46.2% (136/293), and *D. silvarum* carries 2 species of *rickettsia* with a positive rate of 59.09% (13/22). *Haemaphysalis concinna* (*H. concinna*) carries 1 species of *rickettsia*, and the positive rate is 60.00% (3/5). Further studies have shown that the dominant tick species in the Qigan area is *I. persulcatus*, which is also one of the main vectors of tick-borne infectious diseases in the area. Cheng et al. studied the distribution, carrying, and co-infection of ticks with new tick-borne pathogens in 11 areas including Suifenhe, Heilongjiang Province [46]. The results showed that a total of 1306 ticks were collected, including *D. silvarum*, *I. persulcatus*, *H. concinna*, and *Haemaphysalis japonica* (*H. japonica*). A total of 528 cases of *SFGR*, including *R. heilongjiangensis*, *CRT*, and *R. raoultii*, were detected in the collected ticks, with a positive rate of 40.43%. A total of 12 cases of tick-borne pathogen co-infection were detected, and the co-infection rate was 0.92%.

Sun et al. collected *D. nuttalli* from Wusu Tianshan Mountain in Xinjiang to detect *SFGR* in ticks by PCR [47]. The positive rate of *SFGR* among 106 ticks was 57.23%. *SFGR* sequence analysis and phylogenetic tree results showed that they belonged to *R. raoultii* and *R. sibirica*. To investigate the distribution of *SFGR* in ticks in Xinjiang, Qu Zhiqiang et al. collected 2079 free ticks from 10 counties (cities) and parasitic ticks on domestic animals by using the flag method and PCR amplification of ompA gene. A total of six *SFGR* pathogens were detected, namely *R. lauticus*, *R. sibirica*, *R. slovak*, *R. aeschrlichlimannii*, and two strains of unidentified species, one of which is tentatively designated *Candidatus R. barbariae*. The detection rate of ticks in different regions is 40~80%, with an average detection rate of 51.5% [48]. To study the distribution of tick species and the infection of tick-borne Rickettsia in Qinghai Province, Gao Yue et al. collected 1294 ticks in 20 counties and cities and used morphological and molecular biology methods to identify them. The results showed that there were 674 *Haemaphysalis qinghaiensis* (*H. qinghaiensis*), 462 *D. nuttalli*, and 158 *D. silvarum*. Amplification of gltA and ompA genes revealed that the overall infection rate of *Rickettsia* was 3.90%, including *R. slovaca* and *R. raoultii*, and *R. sibirica* [49]. The above results show that *rickettsia* carried by ticks in Inner Mongolia, Heilongjiang, and Xinjiang has complexity and diversity. The results of free ticks and parasitic ticks also show that there is a certain transmission relationship between ticks and animal hosts.

*Rickettsia* first infects endothelial cells and then proliferates in vascular endothelium, leading to vasculitis. The main clinical symptoms are fever, rash, headache, etc. The common feature of most tick-borne *Rickettsia* diseases is the formation of a burn on the bite site. The good ecological environment of Inner Mongolia, Xinjiang, and other places provides suitable living conditions for the vector ticks and host animals of *Rickettsia*. Although great progress has been made in the prevention and control of infectious diseases, emerging infectious diseases are still a new challenge in the above areas. Isolation and identification of emerging tick-borne *Rickettsia* is still a hot spot in *Rickettsia* research. First, the key areas of tick-borne *Rickettsia* diseases in China need to be investigated; secondly, the isolation and identification of pathogens need to be carried out; finally, the importance of *Rickettsia* needs to be increased, meanwhile, the monitoring of tick-borne diseases and prevention are also extremely important.

### 2.4. Spirochaeta

*Spirochetes* are widely distributed, which can cause intestinal diseases in humans and animals, damage various organs and nervous systems, and even die. *Spirochetes* usually exist in mammals and birds, and ticks can obtain the bacteria by sucking the blood of infected animals. Thereafter, the *spirochetes* migrate to salivary glands through the blood cavity of ticks. When the tick sucks blood, the histamine release factor of the tick is up-regulated, which increases the blood flow to the tick bite site, regulates the permeability of the blood vessel, and promotes the tick to suck blood and spread the disease.

*Spirochetes* are divided into five genera, which are *Borrelia*, *Treponema*, *Leptospira*, *Spinale*, and *Spirochetes*. Tick-transmitted *spirochetes* belongs to the genus *Borrelia*. The pathogenic spirochetes are *Borrelia afzelii* (*B. afzelii*), *B*. *garinii*, *B*. *burgdorferi*, *B*. *bavariensis*, *B*. *bissettii*, *B*. *lusitaniae*, *B*. *kurtenbachii*, *B*. *spielmanii* and *B*. *valaisana*, and so on [50].

To explore the infection of *B. burgdorferi* in ticks in the Qiongzhong area of Hainan Province, Zhang and colleagues collected 120 bovine parasitic ticks in the area, followed by identification of tick species and detection of the *B. burgdorferi* carrying rate by ordinary PCR and nested PCR, respectively [51]. The results showed that the collected ticks belonged to *Boophilus microplus* (*B. microplus*), a total of 44 ticks carried *B. burgdorferi* with a positive rate of 36.67% [51]. Tang et al. collected parasitic ticks on the body surface of yak and identified tick species and *Borrelia* species by morphology and nested PCR technology [52]. Among the 818 ticks collected, *Dermacentor everestianus* (*D. everestianus*) accounted for 78.97%, and *H. qinghaiensis* accounted for 21.03%. Notably, *Borrelia* was isolated from *H. qinghaiensis*. Qiu et al. collected more than 1000 ticks in Songfeng Mountain, Harbin, randomly selected 96 dermat ticks, 48 blood ticks, and 48 *I. persulcatus* for PCR identification, followed by sequencing of PCR amplicons [53]. The results showed that the total detection rate was 8.85%, among which *I. persulcatus* was the tick species with the highest detection rate, followed by blood tick and dermatosis ticks. The results of sequencing showed that there were two genotypes of *spirochetes*, namely *B. burgdorferi* and *B. garinii*. Moreover, Duan et al. used nested PCR to detect *B. burgdorferi* on ticks collected in Gengma County of Yunnan province, China, and found that among the 94 ticks collected, 14 positive ticks were detected as *Ixodes ovatus* (*I. ovatus*), and the detection rate was 14.89% [54]. The detected *B. burgdorferi* has 98~99% homology with that of the Chinese genotype *B. burgdorferi* [54]. These results indicate that *B. burgdorferi* mainly exists in Hainan, Sichuan, Harbin, and other places in China, and the intermediate hosts of this bacterium are diverse.

In addition to the areas mentioned above, ticks also have high detection rates in other areas (Table 1).

## 3. Viral Diseases

The tick-borne virus is a virus obtained by sucking the blood of an infected host by ticks, which is transmitted to the host by ticks, resulting in viral diseases of humans and animals. Except for the *African swine fever virus* (*ASFV*), most tick-borne viruses are RNA viruses [72]. The main tick-borne viruses in China include *tick-borne Encephalitis virus* (*TBEV*), *Crimean–Congo hemorrhagic fever virus* (*CCHFV*), and *severe fever with thrombocytopenia syndrome virus* (*SFTSV*), which are mainly distributed in Qinghai, Xinjiang, and other places [73,74,75,76]. More than 80 kinds of tick-borne viruses have been found in 6 families (Table 2) [77].

To investigate the distribution of ticks and the carrying situation of *TBEV* in Heilongjiang Province, China, Wang et al. collected free ticks and parasitic ticks on the surface of animals and detected *TBEV* by fluorescence quantitative PCR [96]. The 3531 ticks collected were mainly *I. persulcatus*, *D. silvarum,* and *H. concinna*. A total of four *TBEV* strains detected were all carried by *I. persulcatus*, and sequence analysis showed that these *TBEV* belonged to the Far East subtype [96]. Jia N et al. isolated the *Jingmen tick virus* (JMTV) in *Amblyomma javanense* (*A. javanense*) and found that the virus can accumulate in the salivary glands of ticks [88]. Further analysis showed that the virus formed a new sub-lineage, which is different from the JMTV previously reported in China [88]. Gong S et al. collected ticks in Northeast China, and found that the main ticks collected were *Haemaphysalis longicornis* (*H. longicornis*) *D. silvarum*, *I. persulcatus,* and *D. nuttalli* [97]. Liu et al. collected 643 parasitic *Hyalomma asiaticum* (*H. asiaticum*) from camels and sheep in Inner Mongolia and detected tick-borne viruses by PCR [98]. As a result, 4 out of 60 tick samples were detected to have the *CCHFV* gene, and the detection rate was 6.7%, among which one virus was hosted by sheep, and the host of the remaining three viruses was the camel. Wu et al. found that *ASFV* transmitted by ticks is species-specific, and only soft ticks of the genus *Ornithodoros* can transmit *ASFV* [99]. The virus is parasitic in the midgut and blood cells of the tick, and the virus particles are tightly combined with red blood cells. After ticks ingest blood, blood cells bound with virus particles are swallowed and infected by digested cells located in the epithelial tissue of the midgut. The virus replicates in the digested cells and then transfers to the reproductive tissues and salivary glands to replicate again. When a tick sucks blood, it can transmit the virus to susceptible animals, such as pigs and wild boar, or through eggs and menstruation. The above results indicate that there are many types of viruses transmitted by ticks. In addition to the above-mentioned tick-borne viruses that have been discovered, we also need to evaluate other viruses carried by ticks to have a deeper understanding of tick-borne viruses.

After the ticks suck the host blood, the virus enters the host cell by endocytosis, replicates in the inner wall of the tick’s midgut, spreads to the hemolymph, infects different tissues, and is excreted from the cell by exocytosis [100]. The diversity of tick-borne viruses leads to uncertainty of ticks and their transmitted pathogens. It is essential to strengthen the prevention and control of tick-borne viruses. In-depth research is needed on the transmission mechanism, characteristics, hosts, virulence, and pathogenicity of tick-borne viruses. To better study tick-borne viruses and prevent the occurrence of tick-borne viruses.

## 4. Parasitic Diseases

Parasitic diseases are also one of the important diseases that endanger the development of the aquaculture industry. They deprive animals of nutrients, leading to slow growth of animals and weight loss. Some parasites migrate in the host body as they decay during the growth and development period, damaging the blood vessels and tissues of the host. The parasites transmitted by ticks are mainly *Babesia* and *Theileriosis* [101,102,103,104].

To detect the infection of *Hepatozoon* sp. in Northeast China, Li et al. collected 2767 ticks, including 168 *D. silvarum*, 212 *D. nuttalli*, 1629 *I. persulcatus*, 378 *Haemaphysalis concinna,* and 380 *Haemaphysalis longicornis*. The results showed that the total infection rate of ticks in northeast China was 1.6% (43/189). The infection rates of *Hepatozoon* sp. were 2.8% (4/12), 1.5% (3/16), 2.0% (29/110), 0.5% (2/26), and 1.1% (5/25) in *D. silvarum*, *D. nuttalli*, *I. persulcatus*, *Haemaphysalis concinna*, and *Haemaphysalis longicornis*, respectively. The results of the evolutionary analysis showed that ticks carried two different genotypes with 99% homology with Japanese mink (Martes melampus melampus). Both could be detected in the ticks of *D*. *silvarum* and *I. persulcatus*, and the infection rate of ticks in different regions was different [105]. To reveal the genetic diversity of *Babesia parvum* and *Theileria orientalis* in southwestern China, Li LH et al. conducted a molecular survey of piriformis in Ixodes ticks in the border counties of China and Myanmar [106]. Animal parasitic ticks and free ticks were collected from Tengchong County, and the tick infection was detected by polymerase chain reaction (PCR). A total of six species of *piriformis* were found, including *Babesia microti* (*B. microti*), *Babesia orientalis* (*B. orientalis*), *Theileria orientalis* (*T. orientalis*), *Theileria luwenshuni* (*T. luwenshuni*), and a newly discovered *Babesia* species named *Babesia Tengchong China*. To understand the pathogens carried by ticks in Shanghai, Zhang and others collected ticks in Shanghai and evaluated the pathogens carried by ticks via nested PCR [107]. The results showed that the collected tick was *Rhipicephalus sanguineus* (*R. sanguineus*), and the protozoan carried by *R. sanguineus* was *Babesia*
*c**anis*, (*B. canis*). In summary, ticks can serve as storage hosts for *Piriformis*, *Babesia*, and *Theileria*, and pose a greater potential threat to human and animal health.

The incidence of parasites is related to the vegetation and altitude in different regions. Areas with low altitude and dense vegetation are more suitable for the growth and reproduction of ticks, thereby affecting the incidence of tick-borne parasitic diseases [103,108,109]. Different feeding methods also have an impact on the incidence of parasitic diseases. Free-range animals are more likely to be exposed to ticks during the grazing process, so the infection rate is higher than that of captive animals [110,111,112]. Appropriately reducing grazing during the months of tick activity and giving animals a good breeding environment can reduce contact with ticks, and to a certain extent, can reduce the incidence of parasitic diseases.

## 5. Detection, Prevention, and Treatment of Tick-Borne Animal Diseases

Early prevention, detection, treatment, as well as comprehensive prevention and control are the keys to controlling tick-borne diseases. Early prevention can greatly reduce economic losses. Elimination of ticks is the key to preventing tick-borne diseases. According to the season of tick activity and the differences of different tick species, planned tick eradication can be carried out. Eliminating transmission vectors is the main way to prevent infection.

With the development of research, more and more methods and strategies are used for tick detection and treatment. However, different methods are used to kill ticks in different regions. In Qinghai, Xinjiang, Gansu, and other areas, some livestock owners chose 0.05% insecticide solution prepared by 0.2% diformamidine emulsion to thoroughly spray in the enclosure, while some livestock owners chose 0.1~0.2% aqueous solution for the medicinal bath to repel insects, which can achieve a better insecticidal effect. Furthermore, some livestock owners choose to inject ivermectin or abamectin subcutaneously to expel ticks and apply waste oil on the surface of livestock to reduce tick bites [113]. Intramuscular injection of ivermectin should be combined with the bodyweight of the calf, and the drug should be injected every 5 d, which can achieve good results and can kill the ticks on the surface of the calf. To control the secondary infection caused by tick-borne pathogens, drug treatment can be used, including glucose injection, sodium bicarbonate injection, normal saline, penicillin sodium, etc. In Hefei, livestock owners chose environmental and biological control methods to reduce the dormant sites of ticks, which had significant effects and were easy to operate [114] However, excessive use of tick-killing drugs can also cause adverse reactions to animals. Therefore, researchers can immunize animals according to the antigens produced by ticks, develop safe and effective vaccines, and formulate different immunizations according to the distribution of different regions and species of ticks.

Furthermore, ticks and tick-borne diseases are a huge challenge to veterinary public health. Mosquito repellents and acaricides are commonly used to control ticks and tick-borne diseases. To protect humans, livestock, and pets from ticks, strategies for vaccine resistance to tick infection are being developed. Zhao et al. found that the serine protease inhibitor (L7LTU1 protein) of *Rhipicephalua* was a feasible candidate vaccine, which is a secreted protein with hydrophilic properties [115]. The protein has similar amino acid sequences in multiple tick species and has good antigenic conserved properties. Song R. et al. used recombinant IFN-γ molecule as an adjuvant of anti-tick vaccine recombinant cathepsin L-like cysteine protease from H. asiaticum tick (rHasCPL) and found that rHasCPL combined with rIFN-γ could induce significant humoral and cellular responses in mice and protect mice from the tick challenge [116]. Huercha et al. cloned and expressed the mu-class glutathione S-transferase (GST) of *Dermacantor marginatus* (rDmGST). The results showed that the engorgement rate, total egg mass, and egg hatching rate of adult female ticks decreased, and the total vaccine efficacy was 43.69% in rDmGST-immunized rabbit challenged with ticks [117].

In addition to conventional PCR detection for ticks and tick-borne diseases, Chang et al. established a dual fluorescence quantitative PCR method for simultaneous diagnosis of Erich’s disease and Lyme disease based on the groEL gene of Erich’s disease and ospA gene of *Borrelia burgdorferi*, respectively. This method could specifically amplify only the genes of *Erich* and *B*. *burgdorferi*, and was negative for *Leptospira*, *Rickettsia*, *Escherichia coli,* and *Pseudomonas aeruginosa*. The detection limit was 1 × 10^1^ copies/μL, and the coefficient of variation of Tm values in and between batches was less than 0.1%, indicating that dual-fluorescence quantitative PCR is superior to ordinary PCR in the detection of Ehrlich disease and Lyme disease [118].

## 6. Conclusions and Perspective

A tick is an external parasite that sucks blood from various wild and domestic animals [119], which can cause direct damage to animals. Since tick-borne diseases can seriously endanger human and animal health, and with the continuous discovery of tick-borne pathogens, emerging tick-borne diseases have increasingly become one of the focuses of infectious diseases. The pathogenic biological characteristics, vector organisms, natural hosts of infection, and endemic areas of emerging tick-borne diseases have become important scientific issues to be solved urgently.

For tick-borne diseases, molecular diagnostic techniques are a common method of detection. However, due to the sudden and sporadic nature of tick-borne infectious diseases, commercial diagnostic kits are relatively single and lag and standard methods are not perfect. Therefore, it is of great importance to establish big data or network of ticks and tick-borne viruses and to form a rapid and high-throughput identification and testing platform for monitoring a variety of tick-borne viruses, which can provide guidance and reference for vector surveillance and pathogen detection.

## Figures and Tables

**Table 1 viruses-14-00355-t001:** Detection rate of tick-borne bacteria.

Species	Tick Species	Areas	Detection Rate (%)	Reference
*Brucella*	*Dermacentor marginatus*	Xinjiang	50	[23]
		Xinjiang	4.6	[22]
	*Dermacentor nuttalli*	Xinjiang	36	[23]
		Inner Mongolia	0~87.8	[24]
	*Hyalomma anatolicum*	Xinjiang	60	[23]
*Tulabacterium*	*Dermacentor nuttalli*	Gansu	6.03	[55]
	*Dermacentor silvarum*	Gansu	9.59	[55]
	*Hyalomma asiaticum*	Xinjiang	100	[24]
	*Haemaphysalis longicornis*	Gansu Liaoning	0.96 54	[55] [56]
	*Haemaphysalis qinghaiensis*	Gansu	6.1	[55]
	*Ixodes persulcatus*	Gansu	1.7	[55]
*Rickettsia*	*Dermacentor marginatus*	Xinjiang	18.44	[57]
	*Dermacentor nuttalli*	Xinjiang	16.92	[57]
		Harbin	4.2	[41]
		Xinjiang	15	[58]
		Inner Mongolia	81.82	[59]
	*Haemaphysalis punctata*	Xinjiang	31.7	[57]
	*Hyalomma asiaticum*	Inner Mongolia	4.7	[60]
	*Rhipicephalus turanicus*	Xinjiang	23.23	[57]
	*Ixodes persulcatus*	Inner Mongolia	69.44	[59]
	*Rhipicephalus microplus*	Xian	40.4	[61]
	*Ixodes kaiseri*	Xinjiang	6	[58]
	*Haemaphysalis longicornis*	Hubei	50	[62]
		Sichuan	25.53	[63]
		Poyanghu	2.31	[64]
		Xian	22.6	[61]
	*Haemaphysalis qinghaiensis*	Harbin	9.7	[41]
		Huaian	3.61	[65]
		Qinghai	9.8	[66]
	*Haemaphysalis flava*	Poyanghu	52.38	[64]
		Sichuan	27.3	[63]
	*Dermacentor silvarum*	Harbin	6	[41]
		Xinjiang	9	[56]
	*Rhipicephalus sanguineus*	Taiwan	2.2	[67]
*Spirochaeta*	*Hyalomma asiaticum*	Inner Mongolia	1.3	[60]
	*Ixodes persulcatus*	Inner Mongolia	2.6	[68]
		Heilongjiang	3	[69]
	*Dermacentor nuttalli*	Inner Mongolia	0.76	[68]
	*Dermacentor silvarum*	Inner Mongolia	1.3	[16]
	*Rhipicephalus sanguineus*	Zhejiang	2.86	[70]
		Shaghai	0.56	
	*Haemaphysalis concinna*	Heilongjiang	2.8	[69]
	*Haemaphysalis longicornis*	Inner Mongolia	0.4	[69]
		Jiangsu	1.2	[71]
		Shanghai	10.13	[70]

**Table 2 viruses-14-00355-t002:** Classification of tick-borne viruses.

Order	Family	Genus	Species	Reference
*Bunyavirales*	*Nairoviridae*	*Orthonairovirus*	*Crimean–Congo hemorrhagic fever virus *, Dera Ghazi Khan virus*, *Dugbe virus*, *Farallon virus*, *Ganjam virus*, *Hughes virus*, *Nairobi sheep disease virus **, *Punta Salinas virus*, *Qalyub virus*, *Sakhalin virus*, *Soldado virus*, *Thiafora virus*	[77,78]
*Bunyavirales*	*Peribunyaviridae*	*Herbevirus* *Orthobunyavirus*	*Bahig virus*, *Matruh virus*	[79,80]
*Bunyavirales*	*Phenuiviridae*	*Banyangvirus* *Goukovirus* *Tenuivirus* *Phasivirus* *Phlebovirus*	*Guertu virus*, *severe fever with thrombocytopenia syndrome virus *, Dabieshan tick virus, Heartland virus*, *Hunter Island virus*, *severe fever with thrombocytopenia syndrome virus **, *Bhanja virus*, *Palma virus*, *Kaisodi virus*, *Khasan virus*, *Lanjan virus*, *Silverwater virus*	[81,82,83]
*Mononegavirales*	*Nyamiviridae*	*Nyavirus* *Peropuvirus* *Socyvirus*	*Midway nyavirus*, *Sierra–Nevada nyavirus*, *Nyamanini nyavirus*	[84]
*Mononegavirales*	*Rhabdoviridae*	*Almendravirus* *Cytorhabdovirus* *Dichorhavirus* *Ephemerovirus* *Lyssavirus* *Novirhabdovirus* *Nucleorhabdovirus* *Perhabdovirus* *Sigmavirus* *Varicosavirus* *Vesiculovirus* *Sprivivirus* *Sripuvirus* *Tibrovirus* *Tupavirus* *Curiovirus* *Hapavirus* *Ledantevirus*	*Barur virus*, *Yongjia tick virus **, *Isfahan virus*, *New Minto virus*, *Sawgrass virus*, *Long Island tick rhabdovirus*, *Zahedan rhabdovirus*, *Connecticut virus*, *Kolente virus*	[85]
*Asfuvirales*	*Asfarviridae*	*Asfivirus*	*African swine fever virus **	[86,87,88]
*Amarillovirales*	*Flaviviridae*	*Flavivirus* *Hepacivirus* *Pestivirus* *Pegivirus*	*Alongshan virus *, Jingmen tick virus *, Kyasanur Forest disease virus*, *Louping ill virus*, *Omsk hemorrhagic fever virus*, *Powassan virus*, *tick-borne encephalitis virus **, *Gadgets Gully virus*, *Karshi virus*, *Langat virus*, *Royal Farm virus*, *Meaban virus*, *Saumarez Reef virus*, *Tyuleniy virus*	[89,90,91,92,93]
*Articulavirales*	*Orthomyxoviridae*	*Quarjavirus* *Thogotovirus*	*Johnston Atoll virus*, *Quaranfil virus*, *Dhori virus*, *Thogoto virus*	[94]
*Reovirales*	*Reoviridae*	*Orbiviruses* *Coltiviruses*	*Bluetongue virus *, Colorado tick fever virus*, *Eyach virus*, *Orbivirus **, *Tibet orbivirus*, *Chenuda virus*, *Essaouira virus, Huacho virus*, *Kala Iris virus*, *Mono Lake virus*, *Sixgun city virus*, *Chobar Gorge virus*, *Great Island virus*, *Kemerovo virus*, *Lipovnik virus*, *Tribec virus*, *Seletar virus*, *Wad Medani virus*, *St Croix River virus*	[71,95]

Note: *, tick-borne viruses isolated in China.

## Data Availability

Not applicable.

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
