# Peer review of "Recent Progress on Tick-Borne Animal Diseases of Veterinary and Public Health Significance in China"

_viruses, 2022, doi:10.3390/v14020355_

Round 1
Reviewer 1 Report
The manuscript reviewed the current epidemics and detection of tick-borne pathogens related to animal diseases in China. The following comments may help to improve the manuscript.
- Figure 1 shows the incidence and distribution of brucellosis. I suggest to better show the differential incidence from high to low levels by gradients of same color.
- Line 72 -73, 79, Please show the copies in correct format.
- Could there be a table to summary the tick species and Brucella, Tulabcterium, Richettsia, or Spirochaeta detection rates in China according to the information of 2.1?
- Line 101 replace “my country” by “China”
- Reference 53 is not about the taxonomy of tic-borne viruses identified in China.
- This review focuses on tick-borne animal diseases in China, and stated the tick-borne viruses in China belong to 2 orders and 9 families. But Table 1 listed 6 families. The number of genera is more than 12, and viruses in the table contains are more than animal disease-related viruses. Some viruses were not reported from China.
- Detection of the tick-borne animal pathogens was elaborated throughout the whole manuscript, which however weakens the description of the healthy importance of these animal diseases to livestock industry or wild environment and potential threats to human health.
- Line 349 program..
- In the perspective, please provide more information about vaccination to animals to protect them from infection of these pathogens. Any detailed progress on the strategy design to kill ticks in the epidemic regions and seasons? Any detailed strategies about improving pathogen detection which could benefit the livestock industry? Better to provide examples of different locations.
Author Response
Dear Editor and Reviewers:
Thank you very much for your letter and the reviewers’ comments concerning our manuscript entitled “Research progress of tick-borne diseases in China” (viruses-1520831).This comments are all valuable and very helpful for revising and improving our paper, as well as the important guiding significance to our researches. We have studied comments carefully and have made correction which we hope meet with approval. Revised portion are marked in blue in the paper. The main corrections in the paper and the responds to the reviewers’ comments are as following:
Responds to the reviewer’s comments:
- Figure 1 shows the incidence and distribution of brucellosis. I suggest to better show the differential incidence from high to low levels by gradients of same color.
Response: We are very sorry that we neglected the problem of color matching of pictures. The pictures have been shown with the same color from light to dark to indicate the incidence and distribution of Brucella.
- Line 72 -73, 79, Please show the copies in correct format.
Response: We are very sorry for our erroneous writing about the numbers. References with numbers have been replaced.
- Could there be a table to summary the tick species and Brucella, Tulabcterium, Richettsia, or Spirochaeta detection rates in China according to the information of 2.1
Response: A table has been added and summarised for the detection rates of Brucella, Tula bacteria, Rickettsia and Spirochetes.
- Line 101 replace “my country” by “China”
Response: We have changed “my country” to “China”.
- Reference 53 is not about the taxonomy of tic-borne viruses identified in China.
Response: Reference 53 is indeed not the classification of tick-borne viruses isolated in China. It is the classification of all tick-borne viruses. Because there is less information in this area, the classification of all tick-borne viruses is selected.
- This review focuses on tick-borne animal diseases in China, and stated the tick-borne viruses in China belong to 2 orders and 9 families. But Table 1 listed 6 families. The number of genera is more than 12, and viruses in the table contains are more than animal disease-related viruses. Some viruses were not reported from China.
Response: Tick-borne viruses do belong to 2 orders and 9 families, but there are 6 families of known tick-borne viruses. It is true that some of these viruses have not been isolated in China, and there are not many references in China on the isolation of tick-borne viruses, so all tick-borne viruses are listed in the table.
- Detection of the tick-borne animal pathogens was elaborated throughout the whole manuscript, which however weakens the description of the healthy importance of these animal diseases to livestock industry or wild environment and potential threats to human health.
Response: Thank you very much for your suggestions. Your suggestions are very helpful for me to revise the article and improve the level of the article. According to your suggestion, the corresponding description has been added at the end of the article.
- In the perspective, please provide more information about vaccination to animals to protect them from infection of these pathogens. Any detailed progress on the strategy design to kill ticks in the epidemic regions and seasons? Any detailed strategies about improving pathogen detection which could benefit the livestock industry? Better to provide examples of different locations.
Response: We appreciate your comments to help us improve the article, and have added a corresponding description at the end of the article.
Thanks again for your suggestion, hope to learn more from you!
Reviewer 2 Report
The submitted manuscript is a review whose main objective is to discuss recent progress on tick-borne animal diseases in China to provide a theoretical basis for the prevention and control of tick-borne diseases.
Several reviews on the topic of the tick-borne disease have been previously reported in the literature, including China, some of which are not cited in the submitted manuscript see, for example:
Wu XB, Na RH, Wei SS, Zhu JS, Peng HJ. (2013). Distribution of tick-borne diseases in China. Parasites & Vectors, 6, 119. https://doi.org/10.1186/1756-3305-6-119
Nelder MP, Russell CB, Sheehan NJ, Sander B, Moore S, Li Y, Johnson S, Patel SN, Sider D. (2016). Human pathogens associated with the black-legged tick Ixodes scapularis: a systematic review. Parasites & Vectors, 9, 265. https://doi.org/10.1186/s13071-016-1529-y
Sanchez-Vicente S, Tagliafierro T, Coleman JL, Benach JL, Tokarz R. (2019). Polymicrobial Nature of Tick-Borne Diseases. mBio, 10(5), e02055-19. https://doi.org/10.1128/mBio.02055-19
Guo WB, Shi WQ, Wang Q, Pan YS, Chang QC, Jiang BG, Cheng JX, Cui XM, Zhou YH, Wei JT, Sun Y, Jiang JF, Jia N, Cao WC. (2021). Distribution of Dermacentor silvarum and Associated Pathogens: Meta-Analysis of Global Published Data and a Field Survey in China. International journal of environmental research and public health, 18(9), 4430. https://doi.org/10.3390/ijerph18094430
Zhao GP, Wang YX, Fan ZW, Ji Y, Liu MJ, Zhang WH, Li XL, Zhou SX, Li H, Liang S, Liu W, Yang Y, Fang LQ. (2021). Mapping ticks and tick-borne pathogens in China. Nature communications, 12(1), 1075. https://doi.org/10.1038/s41467-021-21375-1
Tokarz R, Lipkin WI. (2021). Discovery and Surveillance of Tick-Borne Pathogens. Journal of medical entomology, 58(4), 1525–1535. https://doi.org/10.1093/jme/tjaa269
Vanmechelen B, Merino M, Vergote V, Laenen L, Thijssen M, Martí-Carreras J, Claerebout E, Maes P. (2021). Exploration of the Ixodes ricinus virosphere unveils an extensive virus diversity including novel coltiviruses and other reoviruses. Virus evolution, 7(2), veab066. https://doi.org/10.1093/ve/veab066
The present review pretends to provide an update of the epidemiological status of tick-borne animal diseases in China, with useful information on available tools for the detection, identification, and classification of tick-borne pathogens (Bacteria, Rickettsia, Virus, and Protozoa) of veterinary and zoonotic importance, as well as on the control of ticks. However, important items worth revision were found by the reviewer that authors should revise in order to improve the quality of the submitted manuscript:
Title
The title could be changed to “Recent progress on tick-borne animal diseases of veterinary and public health significance in China” to more precisely identify the study subject to review.
Abstract:
Line 10. Use plural for term ‘Tick’
Introduction
Line 20. Statement ‘Ticks are commonly known as grass crawlers and bovine lice’ needs to be revised, as ticks are not commonly known as bovine lice.
Line 22. Correct statement ‘playing important vectors or intermediates’. Should be ‘playing important roles as vectors or intermediates for…’
Lines 23-24. Check and correct statement ‘Vector, the pathogen carried in its body is second only to mosquitoes’ to make sense.
Line 25. Change statement ‘Ticks belong to the order Acariidae, Ixodoidea,’ for ‘Ticks belong to the class Acariidae, order Ixodoidea,’ and ‘..which consists of Ixodiae, Argasidae, and Nuttalliellidae…’ for ‘…which consists of families Ixodidae, Argasidae, and Nuttalliellidae’
Line 28. Correct term ‘Ixodesidae’
There are a number of citations in this and other sections of the manuscript (see references number 7, 9, 12, 14, 21, 25, 26, 28, 35, 46, 48, 49, 77, 78, 86, 87, 88, 90) that refer to a Master`s, or Ph.D. thesis. However, in the reference section, there is no indication of the URL and/or indication on the date the reference was consulted. Most importantly, these citations are not easily accessed when searched by the reviewer with, for example, the Google search engine. This, most probably because they are in the Chinese language. Thus, it might be an important drawback of the manuscript as researchers from countries other than China might be interested in such references and are unable to have access to them.
Bacterial disease
Line 54. Delete ‘and so on’
Line 60. Can use ‘…prevail…’ instead of ‘…existed…’
Line 67. Change ‘Huang et al.’ for ‘Huang, 2017’ as reference 25 is a single author cite.
Line 72. ‘Brucella’, as a scientific name, should be italicized. Check this issue throughout the manuscript, including the references section, as most of the scientific names are not typed in italics (see, for example, manuscript lines 74, 76, 77, 82, 84, 85,86, 88-110).
Line 72. Use correct scientific notation for ‘1.639 × 101’
Line 73. Use correct scientific notation for ‘2.4 × 105’
Line 75. Change ‘Huang et al.’ for ‘Huang, 2019’ as reference 26 is a single author cite.
Line 80. Use correct scientific notation for ‘7.19×102~1.06×106’
There are a number of published scientific papers on tick-borne bacteria, rickettsiae, viruses, and protozoan parasites in China that were not taken into account to truly update the current situation on tick-borne diseases in China:
Chen Z, Liu Q, Liu JQ, Xu BL, Lv S, Xia S, et al. (2014). Tick-borne pathogens and associated co-infections in ticks collected from domestic animals in central China. Parasites & Vectors. 7:237
Dong, Z., Yang, M., Wang, Z., Zhao, S., Xie, S., Yang, Y., Liu, G., Zhao, S., Xie, J., Liu, Q., & Wang, Y. (2021). Human Tacheng Tick Virus 2 Infection, China, 2019. Emerging infectious diseases, 27(2), 594–598. https://doi.org/10.3201/eid2702.191486
Guo, WP., Xie, GC., Li, D. et al. Molecular detection and genetic characteristics of Babesia gibsoni in dogs in Shaanxi Province, China. Parasites Vectors 13, 366 (2020). https://doi.org/10.1186/s13071-020-04232-w
Guo, J. J., Lin, X. D., Chen, Y. M., Hao, Z. Y., Wang, Z. X., Yu, Z. M., Lu, M., Li, K., Qin, X. C., Wang, W., Holmes, E. C., Hou, W., & Zhang, Y. Z. (2020). Diversity and circulation of Jingmen tick virus in ticks and mammals. Virus evolution, 6(2), veaa051. https://doi.org/10.1093/ve/veaa051
Guo, W. B., Shi, W. Q., Wang, Q., Pan, Y. S., Chang, Q. C., Jiang, B. G., Cheng, J. X., Cui, X. M., Zhou, Y. H., Wei, J. T., Sun, Y., Jiang, J. F., Jia, N., & Cao, W. C. (2021). Distribution of Dermacentor silvarum and Associated Pathogens: Meta-Analysis of Global Published Data and a Field Survey in China. International journal of environmental research and public health, 18(9), 4430. https://doi.org/10.3390/ijerph18094430
Jiao J, Zhang J, He P, OuYang X, Yu Y, Wen B, Sun Y, Yuan Q, Xiong X. (2021). Identification of Tick-Borne Pathogens and Genotyping of Coxiella burnetii in Rhipicephalus microplus in Yunnan Province, China. (2021). Frontiers in Microbiology, 12, 2718. https://www.frontiersin.org/article/10.3389/fmicb.2021.736484
Li LH, Wang JZ, Zhu D, Li XS, Lu Y, Yin SQ, Li SG, Zhang Y, Zhou XN. (2020). Detection of novel piroplasmid species and Babesia microti and Theileria orientalis genotypes in hard ticks from Tengchong County, Southwest China. Parasitol Res. 119(4):1259-1269. doi: 10.1007/s00436-020-06622-6.
Li Y, Wen X, Li M, Moumouni PFA, Galon EM, Guo Q, Rizk MA, Liu M, Li J, Ji S, Tumwebaze MA, Byamukama B, Chahan B, Xuan X. (2020). Molecular detection of tick-borne pathogens harbored by ticks collected from livestock in the Xinjiang Uygur Autonomous Region, China. Ticks Tick Borne Dis. 11(5):101478. doi: 10.1016/j.ttbdis.2020.101478.
Liu X, Zhang X, Wang Z, Dong Z, Xie S, Jiang M, Song R, Ma J, Chen S, Chen K, Zhang H, Si X, Li C, Jin N, Wang Y, Liu Q. (2020). A Tentative Tamdy Orthonairovirus Related to Febrile Illness in Northwestern China. Clin Infect Dis. 6;70(10):2155-2160. doi: 10.1093/cid/ciz602.
Meng F, Ding M, Tan Z, Zhao Z, Xu L, Wu J, He B, Tu C. 2019. Virome analysis of tick-borne viruses in Heilongjiang Province, China. Ticks Tick Borne Dis. 10(2):412-420. doi: 10.1016/j.ttbdis.2018.12.002.
Shao JW, Guo LY, Yuan YX, Ma J, Chen JM, Liu Q. (2021). A Novel Subtype of Bovine Hepacivirus Identified in Ticks Reveals the Genetic Diversity and Evolution of Bovine Hepacivirus. Viruses, 13(11), 2206. https://doi.org/10.3390/v13112206
Song R, Wang Q, Guo F, Liu X, Song S, Chen C, et al. (2018). Detection of Babesia spp., Theileria spp. and Anaplasma ovis in Border Regions, northwestern China. Transbound Emerg Dis. 65:1537–44.
Wang A, Pang Z, Liu L, Ma Q, Han Y, Guan Z, Qin H, Niu G. (2019). Detection and Phylogenetic Analysis of a Novel Tick-Borne Virus in Yunnan and Guizhou Provinces, Southwestern China. Pathogens. 10(9):1143. doi: 10.3390/pathogens10091143.
Wang J, Liu J, Yang J, Wang X, Li Z, Jianlin X, et al. (2019). The first molecular detection and genetic diversity of Babesia caballi and Theileria equi in horses of Gansu province, China. Ticks Tick Borne Dis. 10:528–32.
Wang Q, Zhao S, Wureli H, Xie S, Chen C, Wei Q, Cui B, Tu C, Wang Y. (2018). Brucella melitensis and B. abortus in eggs, larvae and engorged females of Dermacentor marginatus. Ticks Tick Borne Dis. 9(4):1045-1048. doi: 10.1016/j.ttbdis.2018.03.021.
Wang ZD, Wang W, Wang NN, Qiu K, Zhang X, Tana G, Liu Q, Zhu XQ. (2019). Prevalence of the emerging novel Alongshan virus infection in sheep and cattle in Inner Mongolia, northeastern China. Parasites & vectors, 12(1), 450. https://doi.org/10.1186/s13071-019-3707-1
Wang ZD, Wang B, Wei F, Han SZ, Zhang L, Yang ZT, Yan Y, Lv XL, Li L, Wang SC, Song MX, Zhang HJ, Huang SJ, Chen J, Huang FQ, Li S, Liu HH, Hong J, Jin YL, Wang W, Zhou JY, Liu Q. (2019). A New Segmented Virus Associated with Human Febrile Illness in China. N Engl J Med. 30;380(22):2116-2125. doi: 10.1056/NEJMoa1805068.
Wu XB, Na RH, Wei SS, Zhu JS, Peng HJ. (2013). Distribution of tick-borne diseases in China. Parasites & vectors, 6, 119. https://doi.org/10.1186/1756-3305-6-119
Xu L, Guo M, Hu B, Zhou H, Yang W, Hui L, Huang R, Zhan J, Shi W, Wu Y. (2021). Tick virome diversity in Hubei Province, China, and the influence of host ecology. Virus Evolution, 7, 2, veab089, https://doi.org/10.1093/ve/veab089
Wang YC, Wei Z, Lv X, Han S, Wang Z, Fan C, Zhang X, Shao J, Zhao YH, Sui L, Chen C, Liao M, Wang B, Jin N, Li C, Ma J, Hou ZJ, Yang Z, Han Z, Zhang Y, Niu J, Wang W, Wang Y, Liu Q. (2021). A new nairo-like virus associated with human febrile illness in China. Emerging Microbes & Infections, 10:1, 1200-1208. DOI: 10.1080/22221751.2021.1936197
Yu PF, Niu QL, Liu ZJ, Yang JF, Chen Z, Guan GQ, Liu GY, Luo JX, Yin H. (2016). Molecular epidemiological surveillance to assess emergence and re-emergence of tick-borne infections in tick samples from China evaluated by nested PCRs. Acta Tropica, 158, 2016, 181-188, https://doi.org/10.1016/j.actatropica.2016.02.027.
Yu ZM, Chen JT, Qin J, Guo JJ, Li K, Xu QY, Wang W, Lu M, Qin XC, Zhang YZ. (2020). Identification and characterization of Jingmen tick virus in rodents from Xinjiang, China. Infection, Genetics and Evolution, 84, 104411. https://doi.org/10.1016/j.meegid.2020.104411
Zhang K, Li A, Wang Y, Zhang J, Chen Y, Wang H, Shi R, Qiu Y. (2021). Investigation of the presence of Ochrobactrum spp. and Brucella spp. in Haemaphysalis longicornis. Ticks and Tick-borne Diseases, 12, 1, 101588, https://doi.org/10.1016/j.ttbdis.2020.101588.
Zhang Y, Hu B, Agwanda B, Fang Y, Wang J, Kuria S, Yang J, Masika M, Tang S, Lichoti J, Fan Z, Shi Z, Ommeh S, Wang H, Deng F, Shen S. (2021). Viromes and surveys of RNA viruses in camel-derived ticks revealing transmission patterns of novel tick-borne viral pathogens in Kenya. Emerging microbes & infections, 10(1), 1975–1987. https://doi.org/10.1080/22221751.2021.1986428
Zhang Y, Jiang L, Yang Y, Xie S, Yuan W, Wang Y. (2021). A tick bite patient with fever and meningitis co-infected with Rickettsia raoultii and Tacheng tick virus 1: a case report. BMC infectious diseases, 21(1), 1187. https://doi.org/10.1186/s12879-021-06877-z
Author Response
Dear Editor and Reviewers:
Thank you very much for your letter and the reviewers’ comments concerning our manuscript entitled “Research progress of tick-borne diseases in China” (viruses-1520831).This comments are all valuable and very helpful for revising and improving our paper, as well as the important guiding significance to our researches. We have studied comments carefully and have made correction which we hope meet with approval. Revised portion are marked in blue in the paper. The main corrections in the paper and the responds to the reviewers’ comments are as following:
Responds to the reviewer’s comments:
- The title could be changed to “Recent progress on tick-borne animal diseases of veterinary and public health significance in China” to more precisely identify the study subject to review.
Response: Thank you for your valuable comments, the title has been changed to “Recent progress on tick-borne animal diseases of veterinary and public health significance in China”.
- Line 10. Use plural for term ‘Tick’.
Response: Thank you for your valuable comments. The line 10 already uses Ticks.
- Line 20. Statement ‘Ticks are commonly known as grass crawlers and bovine lice’ needs to be revised, as ticks are not commonly known as bovine lice.
Response: Thank you for your valuable comments. Line 20 has been modified.
- Line 22. Correct statement ‘playing important vectors or intermediates’. Should be ‘playing important roles as vectors or intermediates for…
Response: Thank you for your valuable comments. Line 20 has been modified.
- R Lines 23-24. Check and correct statement ‘Vector, the pathogen carried in its body is second only to mosquitoes’ to make sense.
Response: Thank you for your valuable comments. Line 23-24 has been modified.
- Line 25. Change statement ‘Ticks belong to the order Acariidae, Ixodoidea,’ for ‘Ticks belong to the class Acariidae, order Ixodoidea,’ and ‘..which consists of Ixodiae, Argasidae, and Nuttalliellidae…’ for ‘…which consists of families Ixodidae, Argasidae, and Nuttalliellidae’.
Response: Thank you for your valuable comments. Line 25 has been modified.
- Line 28. Correct term ‘Ixodesidae’
Response: Thank you for your valuable comments. Line 25 has been modified.
- There are a number of citations in this and other sections of the manuscript (see references number 7, 9, 12, 14, 21, 25, 26, 28, 35, 46, 48, 49, 77, 78, 86, 87, 88, 90) that refer to a Master`s, or Ph.D. thesis. However, in the reference section, there is no indication of the URL and/or indication on the date the reference was consulted. Most importantly, these citations are not easily accessed when searched by the reviewer with, for example, the Google search engine. This, most probably because they are in the Chinese language. Thus, it might be an important drawback of the manuscript as researchers from countries other than China might be interested in such references and are unable to have access to them.
Response: Thank you for your valuable comments . References 7, 9, 12, 14, 21, 25, 26, 28, 35, 46, 48, 49, 77, 78, 86, 87, 88, 90 have been replaced.
- Line 54. Delete ‘and so on’
Response: Thank you for your valuable comments. Line 54has been modified.
- Line 60. Can use ‘…prevail…’ instead of ‘…existed…’
Response: Thank you for your valuable comments. Line 60 has been modified.
- Line 67. Change ‘Huang et al.’ for ‘Huang, 2017’ as reference 25 is a single author cite.
Response: Thank you for your valuable comments. Line 67 has been modified.
- Line 72. ‘Brucella’, as a scientific name, should be italicized. Check this issue throughout the manuscript, including the references section, as most of the scientific names are not typed in italics (see, for example, manuscript lines 74, 76, 77, 82, 84, 85,86, 88-110)
Response: Thank you for your valuable comments. Line 72 has been modified.
- Line 72. Use correct scientific notation for ‘1.639 × 101’
Response: Thank you for your valuable comments. Line 72 has been modified.
- Line 73. Use correct scientific notation for ‘2.4 × 105’
Response: Thank you for your valuable comments. Line 73 has been modified.
- Line 75. Change ‘Huang et al.’ for ‘Huang, 2019’ as reference 26 is a single author cite.
Response: Thank you for your valuable comments. Line 75 has been modified.
- Line 80. Use correct scientific notation for ‘7.19×102~1.06×106’
Response: Thank you for your valuable comments. Line 80 has been modified.
Thank you for the latest references on tick-borne diseases, they have helped me a lot.
Thanks again for your suggestion, hope to learn more from you!
Round 2
Reviewer 1 Report
The authors clarified most of problems and questions. There are still a few of minor points which need to be addressed.
There are reviews about tick-borne viruses which summarized the information of viruses found from China or from other countries. Since the title of this article is about tick-borne animal diseases in China, I insist to present the viral pathogens related to animals in Table 2 or to distinguish the viruses found in China from those found in other countries in this Table by using some markers. Then, this could help to better focus on the topic of the current paper.
In the conclusion and perspective section, the authors added information about vaccines and control strategies without any references. Please add proper ones as indicated in this part.
Author Response
Dear Editors and Reviewers:
Thank you very much for your letter and the reviewers’ comments concerning our manuscript entitled viruses-1520831.This comments are all valuable and very helpful for revising and improving our paper, as well as the important guiding significance to our researches. We have studied comments carefully and have made correction which we hope meet with approval. Revised portion are marked in red in the paper. The main corrections in the paper and the responds to the reviewers’ comments are as following:
Responds to the review·er’s comments:
- Thank you for your suggestion.Viruses that have been isolated in China are marked with "*" in Table 2.
- Thanks for your suggestion, your suggestion helped me a lot. In the fifth part of the article, relevant content about drug treatment, vaccines and detection methods has been added.
Your suggestions are of great help to the improvement of my article, I hope I can learn more from you!
Reviewer 2 Report
The submitted manuscript is a review whose main objective is to discuss recent progress on tick-borne animal diseases in China to provide a theoretical basis for the prevention and control of tick-borne diseases.
Several reviews on the topic of the tick-borne disease have been previously reported in the literature, including China, which were not taken into account to truly update the current situation on tick-borne diseases in China:
Wu XB, Na RH, Wei SS, Zhu JS, Peng HJ. (2013). Distribution of tick-borne diseases in China. Parasites & vectors, 6, 119. https://doi.org/10.1186/1756-3305-6-119
Guo, W. B., Shi, W. Q., Wang, Q., Pan, Y. S., Chang, Q. C., Jiang, B. G., Cheng, J. X., Cui, X. M., Zhou, Y. H., Wei, J. T., Sun, Y., Jiang, J. F., Jia, N., & Cao, W. C. (2021). Distribution of Dermacentor silvarum and Associated Pathogens: Meta-Analysis of Global Published Data and a Field Survey in China. International journal of environmental research and public health, 18(9), 4430. https://doi.org/10.3390/ijerph18094430
Zhao, G. P., Wang, Y. X., Fan, Z. W., Ji, Y., Liu, M. J., Zhang, W. H., Li, X. L., Zhou, S. X., Li, H., Liang, S., Liu, W., Yang, Y., & Fang, L. Q. (2021). Mapping ticks and tick-borne pathogens in China. Nature communications, 12(1), 1075. https://doi.org/10.1038/s41467-021-21375-1
The present review pretends to provide an update of the epidemiological status of tick-borne animal diseases in China, with useful information on available tools for the detection, identification, and classification of tick-borne pathogens (Bacteria, Rickettsia, Virus, and Protozoa) of veterinary and zoonotic importance, as well as on the control of ticks.
Most of the changes suggested by the reviewer have been taken care of by the authors. However, there still are some important items worth revision:
Line 28. Correct term ‘Ixodesidae’ to ‘Ixodidae’
Line 207. ‘Borrelia’, as a scientific name, should be italicized. Check this issue throughout the manuscript, including the references section, as several of the scientific names are not typed in italics (see, for example, manuscript lines 207, 208-210). Check also lines 294, 296 for scientific names, and in several article titles in references section (see for example references 2, 23, 30, 32-38, 43, 44, 47, 52-54, 56-65, 68-71, 73, 78
Line 209. Use correct species name for Borrelia burgdorferi (see also lines 212, 214, 216, 226, 227, 230, 231)
Author Response
Dear Editor and Reviewers:
Thank you very much for your letter and the reviewers’ comments concerning our manuscript entitled “Research progress of tick-borne diseases in China” (viruses-1520831).This comments are all valuable and very helpful for revising and improving our paper, as well as the important guiding significance to our researches. We have studied comments carefully and have made correction which we hope meet with approval. Revised portion are marked in red in the paper. The main corrections in the paper and the responds to the reviewers’ comments are as following:
Responds to the reviewer’s comments:
- Thanks for your suggestion, your suggestion helped me a lot. Lines 28, 207, 208-210, 212, 214, 216, 226, 227, 230, 231, 294, 296 and references 2, 23, 30, 32-38, 43, 44, 47, 52 -54, 56-65, 68-71, 73, 78 were revised.
Thank you for the latest references on tick-borne diseases, they have helped me a lot.
Thanks again for your suggestion, hope to learn more from you!